# Wind Effects for Floating Algae Dynamics in Eutrophic Lakes

Yuchao Zhang [1], Steven Loiselle [2], Kun Shi [3,4,*], Tao Han [3], Min Zhang [3], Minqi Hu [1,5], Yuanyuan Jing [1,5], Lai Lai [1,5] and Pengfei Zhan [1,5]

1 Key Laboratory of Watershed Geographic Sciences, Nanjing Institute of Geography and Limnology, Chinese Academy of Sciences, Nanjing 210008, China; yczhang@niglas.ac.cn (Y.Z.); huminqi16@mails.ucas.ac.cn (M.H.); jingyuanyuan18@mails.ucas.ac.cn (Y.J.); lailai19@mails.ucas.ac.cn (L.L.); zhanpengfei17@mails.ucas.ac.cn (P.Z.)
2 Dipartimento di Biotecnologie, Chimica e Farmacia, University of Siena, CSGI, Via Aldo Moro 2, 53100 Siena, Italy; loiselle@unisi.it
3 State Key Laboratory of Lake Science and Environment, Nanjing Institute of Geography and Limnology, Chinese Academy of Sciences, Nanjing 210008, China; than@niglas.ac.cn (T.H.); mzhang@niglas.ac.cn (M.Z.)
4 Center for Excellence in Tibetan Plateau Earth Sciences, Chinese Academy of Sciences, Beijing 100101, China
5 University of Chinese Academy of Sciences, Beijing 100049, China
* Correspondence: kshi@niglas.ac.cn; Tel.: +86-1826-000-2994

**Abstract:** Wind-speed decline is an important impact of climate change on the eastern Asian atmospheric circulation. Although wind does not determine algae biomass in eutrophic lakes, it is a decisive factor in the formation and severity of algae blooms. Based on 2000–2018 MODIS images, this study compared the effects of wind speed on algal blooms in three typical eutrophic lakes in China: Lake Taihu, Lake Chaohu and Lake Dianchi. The results indicate that climate change has different effects on the wind speed of the three lakes, but a common effect on the vertical distribution of algae. A wind speed of 3.0 m/s was identified as the critical threshold in the vertical distribution of chlorophyll-a concentrations in the three study lakes. The basic characteristics of the periodic variation of wind speed were different, but there was a significant negative correlation between wind speed and floating algal bloom area in all three lakes. In addition, considering lake bathymetry, wind direction could be used to identify locations that were particularly susceptible to algae blooms. We estimated that algal bloom conditions will worsen in the coming decades due to the continuous decline of wind, especially in Lake Taihu, even though the provincial and national governments have made major efforts to reduce eutrophication drivers and restore lake conditions. These results suggest that early warning systems should include a wind-speed threshold of 3.0 m/s to improve control and mitigation of algal blooms on these intensively utilized lakes.

**Keywords:** wind speed; wind direction; floating algae; eutrophic lakes; climate change; phenological processes



## 1. Introduction

Harmful algae blooms (HABs), usually associated with floating cyanobacteria, have international consequences on the availability of freshwater resources, with implications on health and economic activities [1,2]. The growing incidence of HABs has been associated with direct changes in nutrient fluxes and nutrient availability in lakes and other receiving waterbodies. Climate change has a number of indirect influences on nutrient fluxes, as changes in rainfall and temperature influence nutrient retention in surrounding sediment and vegetation and have influenced land use and land cover in many lake catchments. However, direct drivers of algal bloom formation, extension and permanence are influenced by changes to key local meteorological conditions, including air temperature (thermal stratification, primary and secondary production), precipitation (dilution and reduced photosynthetic available solar radiation) and wind speed (thermal stratification, sediment and nutrient upwelling, horizontal lake mixing) [3–5]. Wind-influenced processes link

meteorology, lake circulation and lake bathymetry through stratification, nutrient dynamics and trophic states [6–8]. Recent studies indicate that climate change is expected to reduce wind speed in the eastern Asian atmospheric circulation [9]. In deep lakes, a decreased wind speed favors an earlier onset or longer duration of lake stratification, with impacts on benthic primary and secondary production, with consequences on the nutrient exchange with surface waters [10–13]. In shallow lakes, a decreased wind speed also favors lake stratification, but the influence on nutrient exchange is less clear, depending on optical depth and secondary benthic production. Surface and vertical distributions of the total algae in shallow lakes were markedly affected by wind, and the coefficients of variation of pigment concentrations throughout the water column show a negative correlation with increasing wind [14–16].

In either deep or shallow lakes, the link between wind regime and HABs dynamics can be defined and used for management purposes. These may include the development of early warning systems or the changes in water withdrawal based on predicted or actual meteorological conditions.

In the present study, we use meteorological information and Earth observation data to develop a common early warning approach for three important lakes in China: Lake Taihu, Lake Chaohu and Lake Dianchi. These shallow lakes provide freshwater and important ecological services to the 25 million people in their catchments (Figure 1). Lake and basin characteristics vary among lakes (Table 1), but similar challenges are faced by all three lake authorities in dealing with frequent and large-scale algal blooms [3,17–19] dominated by harmful cyanobacteria [20–22]. We explore the relationship between changing wind conditions on algal bloom intensity and spatial distribution also to understand the impact of climate change on inland waterbody processes.

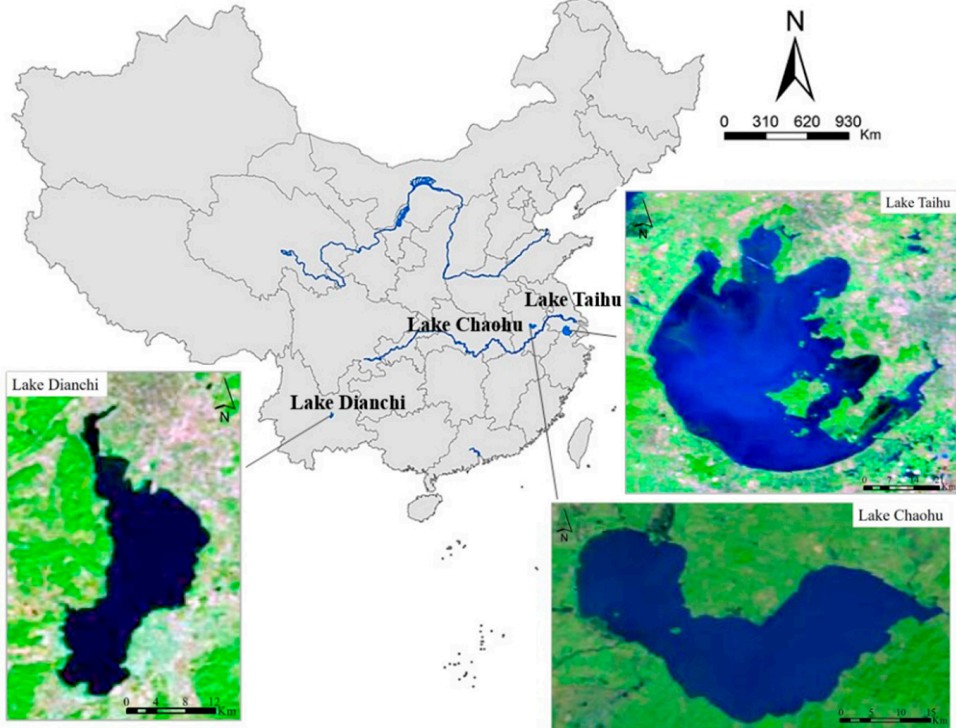

**Figure 1.** The locations of Lake Taihu, Lake Chaohu and Lake Dianchi in China.

**Table 1.** Description of the basic features of Lake Taihu, Lake Chaohu, and Lake Dianchi during 2000–2018.

|  | Lake Taihu | Lake Chaohu | Lake Dianchi |
|---|---|---|---|
| Longitude | 119°55′–120°59′E | 117°17′–117°51′E | 102°35′–102°47′E |
| Latitude | 30°90′–31°54′N | 31°25′–31°43′N | 24°40′–25°02′N |
| Total area of its basin (km$^2$) | 36,900 | 12,938 | 2920 |
| Total water area (km$^2$) | 2338 | 770 | 330 |
| Annual average water depth (m) | 1.9 | 3.0 | 5.0 |
| TN(mg/L) | 2.03–3.71 | 0.54–4.53 | 0.96–1.92 |
| TP(mg/L) | 0.085–0.136 | 0.037–0.292 | 0.021–0.195 |
| Eutrophication state [a] | Moderate | Mild | Mild |
| Climate type | Southeast monsoon | Southeast monsoon | Southwest monsoon |
| Annual average air temperature (°C) | 17.6 | 16.7 | 16.0 |
| Prevailing wind direction | East-southeast | East and south | West-southwest |
| Annual average wind speed (m/s) | 2.9 | 2.4 | 2.3 |

[a] China Environmental Quality Bulletin 2019, Ministry of Ecology and Environment of China.

## 2. Data and Methods

### 2.1. Algal Bloom Monitoring

The Moderate Resolution Imaging Spectroradiometer (MODIS) data were downloaded from the National Aeronautics and Space Administration (NASA) EOS Data Gateway (EDG) (http://modis.gsfc.nasa.gov/, accessed on 8 January 2021). This study used a total of 4800 MODIS images from 2000–2018, all of which were first georeferenced to Universal Transverse Mercator (UTM) projection with an error of less than 0.5 pixel. The resolutions of the different bands were resampled uniformly to 250 m resolution (to match the resolution of the waveband centered at 645 nm). Rayleigh-corrected reflectance (Rrc) was obtained by removing the molecular (Rayleigh) scattering effects [23]. To avoid cloud-induced bias in the area statistics of the floating algae blooms, the cloud-covered areas were masked during Rayleigh correction. In this study, floating algae area monitoring was achieved by an algae pixel-growing algorithm (APA) [17], based on the Floating Algal Index (FAI) [24]. The subpixel floating algae area was obtained by calculating the algae coverage in a pixel, which is a very suitable method for the coarse spatial resolution of MODIS data in practice. APA has been practiced in monitoring floating algae area in Lake Taihu, Lake Chaohu and many other lakes.

### 2.2. Wind Dynamics

Daily data (air temperature, wind direction, wind speed, precipitation, sunshine hours, etc.) were obtained for the nearest national weather stations (i.e., Dongshan, Hefei and Kunming Stations) from 2000 to 2018 from the China Meteorological Data Sharing Service System (http://cdc.cma.gov.cn/home.do, accessed on 8 January 2021).

The Princeton Ocean Model [25–27] was used to determine wind/lake conditions, including the velocity components *u*, *v*, and *w* under different wind speeds and directions. The divergence value was calculated by:

$$div A = \frac{\partial u}{\partial x} + \frac{\partial v}{\partial y} \tag{1}$$

using the velocity components in the x- and y-directions, respectively. If the divergence value at a control volume is larger than 0, the volume is divergent, which indicated the algal bloom is difficult to form; on the contrary, it indicates a convergent volume, and the algae does not want to leave the control volume. The convergence zones of the algae blooms were then identified [28].

*2.3. Long Term and Seasonal Trends Analysis*

A Breaks For Additive Seasonal and Trend (BFAST) was used to decompose each time series, algal blooms and wind dynamics into trend, seasonal and residual (irregular) components [29]. BFAST has been used successfully to estimate phenological changes in ecosystems using time series satellite images [30–33]. We compared seasonal trends of the area of algal blooms and wind speed using the BFAST package for R (developed by R Development Core Team 2009) from Comprehensive R Archive Network (CRAN) (http://CRAN.R-project.org/package=bfast, accessed on 8 January 2021).

## 3. Results

In the 2000–2018 study, Lake Taihu was dominated by southeast–east, northwest–north and northwest winds (43.1%). The prevailing wind in Lake Chaohu was east and south (39.8%), while southwest–west and southwest winds dominated Lake Dianchi (40.7%). The average annual wind speed of Lake Taihu was the largest (2.9 m/s), followed by Lake Chaohu (2.4 m/s). The average wind speed of Lake Dianchi was 2.3 m/s. Lake Dianchi varied the most in wind direction (Coefficient of variation (CV) = 0.19), followed by Lake Taihu (CV = 0.14) and then Lake Chaohu (CV = 0.099).

Wind speed (2000–2018) showed short- and long-term trends for each lake Figure 3. In particular, there was a clear long-term negative trend in Lake Taihu based on wind speed data from 1956. Extending the interannual decomposed linear trend to 2030, the expected annual average wind speed showed as a 14.8% reduction for Lake Taihu from 2.65 to 2.25 m/s. Lakes Chaohu did also show similar clear negative trends in wind speed, but Dianchi did not. The wind speed of Lakes Taihu and Chaohu showed a negative trend in the recent period of increasing algal blooms (Figure 2; Figure 3), while Lake Dianchi showed a general increase.

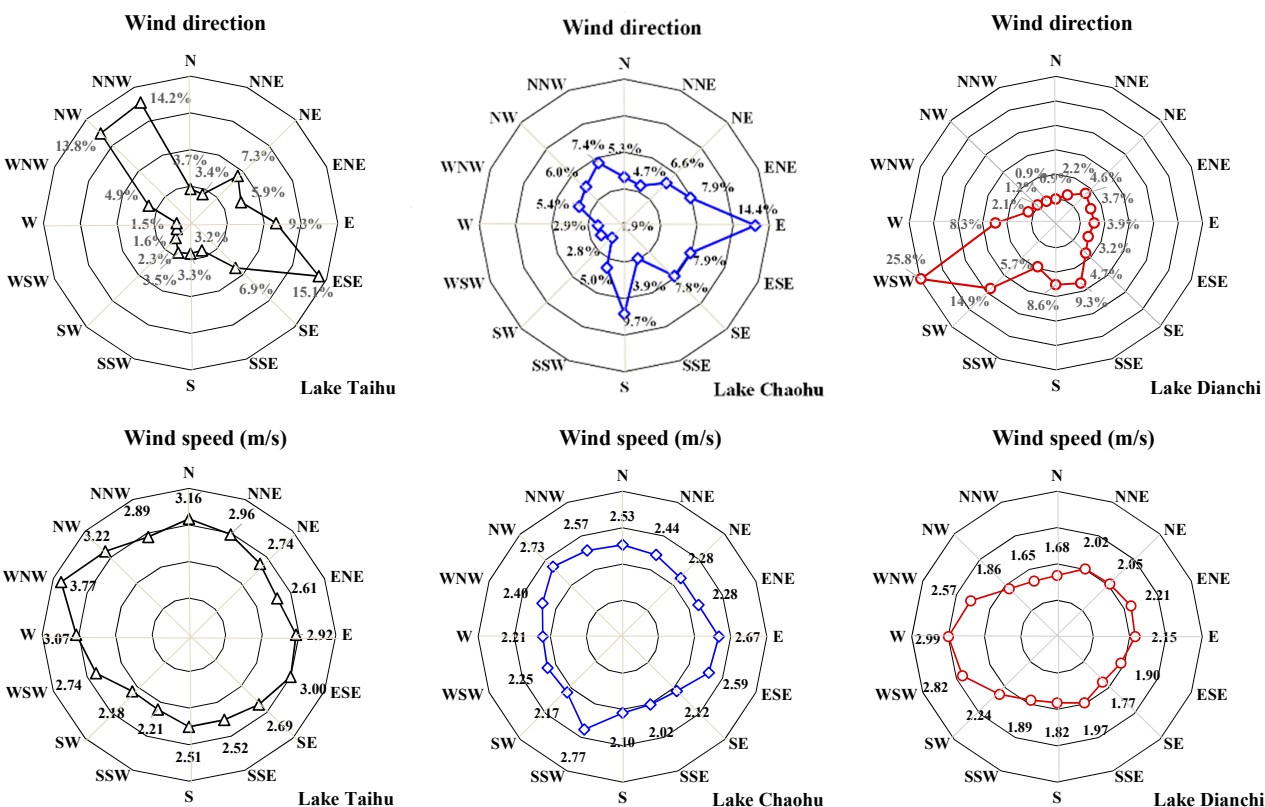

**Figure 2.** Wind speed and directions in Lake Taihu, Lake Chaohu, and Lake Dianchi (2000–2018).

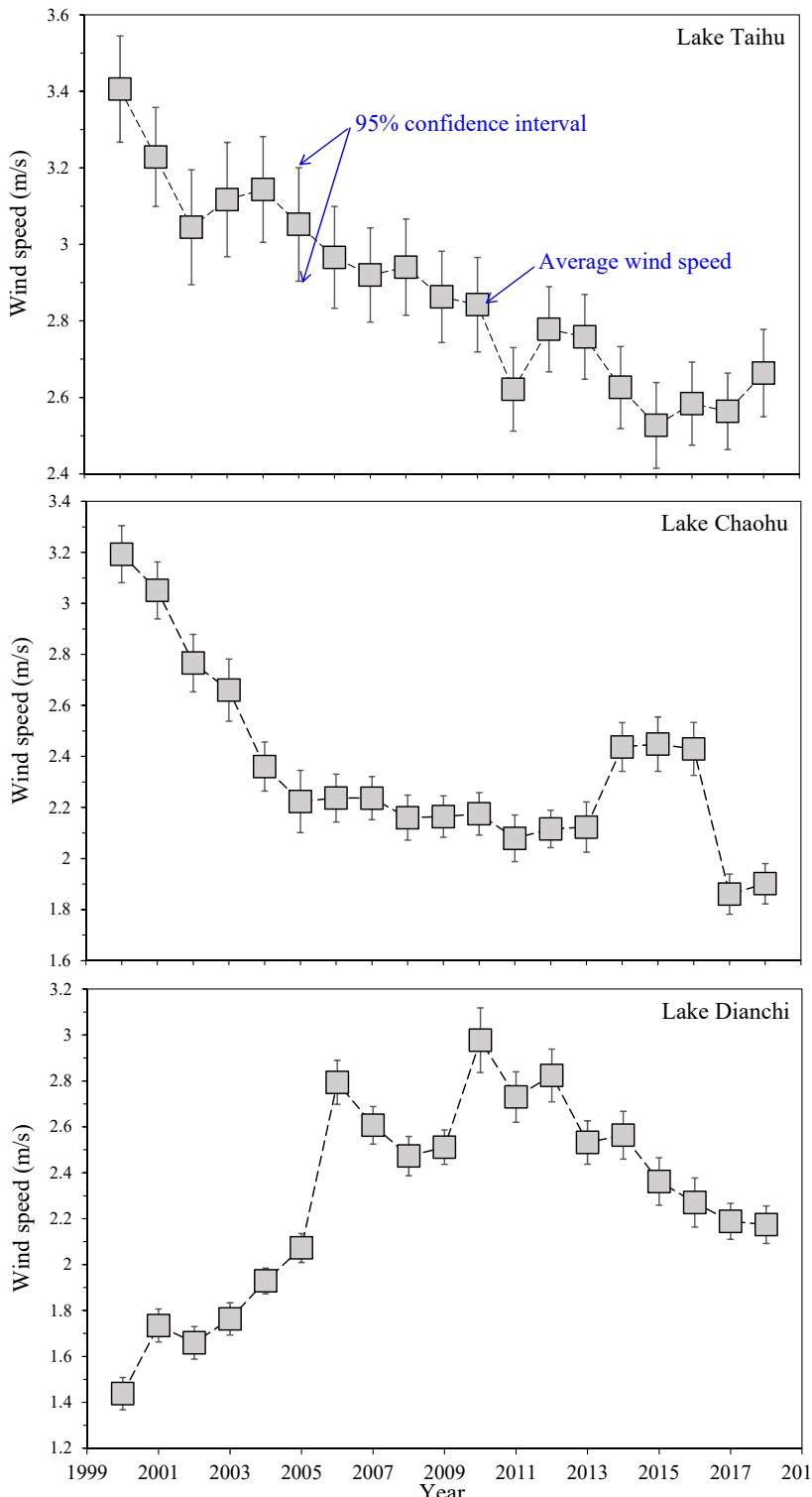

**Figure 3.** Wind speed variations in Lake Taihu, Lake Chaohu and Lake Dianchi from 2000 to 2018.

Monthly algal bloom dynamics showed a seasonal cycle of blooms, with nonlinear interannual trend (Figure 4). The maximum lake surface area of algal blooms for each month in each lake was used to characterize algal bloom dynamics. To normalize the data for interlake comparison, we used the ratio of algal bloom area to the total water area. If the ratio was greater than 20%, the lake was defined as undergoing severe algal blooms. The limit of 20% was selected as lake management authorities due to the top 5% largest algal bloom area in these three lakes.

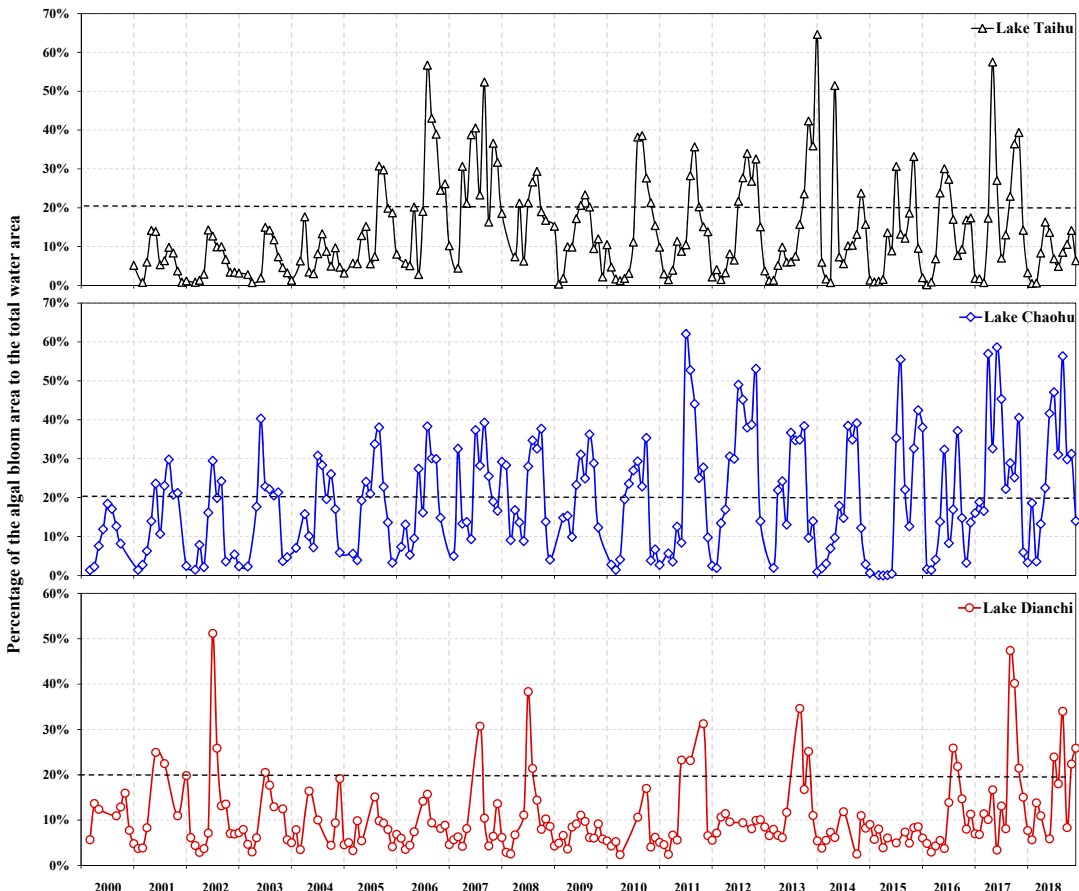

**Figure 4.** Percentage of the algal bloom area to the total water area in Lake Taihu, Lake Chaohu and Lake Dianchi from 2000 to 2018.

Conditions of severe algal blooms occurred in 40% of the total months for Lake Chaohu, 25% for Lake Taihu, and 9% in Lake Dianchi from 2000 to 2018 (Figure 4) and typically in the months of May–August in Lakes Chaohu and Taihu, and in the months of July–December in Lake Dianchi.

Monthly wind speed and algal bloom area showed the expected negative relationship (Figures 5 and 6). The area of algal blooms doubled with a decrease in wind speed from 3 to 1 m/s. The linear relationship between wind speed and bloom area from 1 to 3 m/s was significant and similar among lakes (r = 0.928~0.962, *p* < 0.01). The Pearson coefficient for Lake Taihu showed the stronger sensitivity of this lake to wind speed. This could be associated with its shallowness, eutrophication and dish shape with respect to dominating wind direction [34].

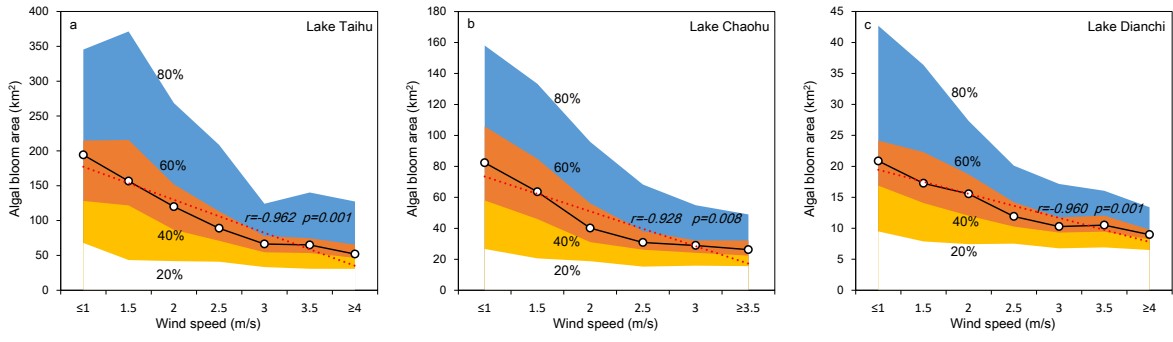

**Figure 5.** Wind characteristics over: (**a**) Lake Taihu; (**b**) Lake Chaohu; and (**c**) Lake Dianchi with respect to the area of floating algal blooms, (20–40% in yellow, 40–60% in orange, and 60–80% in blue), versus the median of the trends of the area of floating algal blooms.

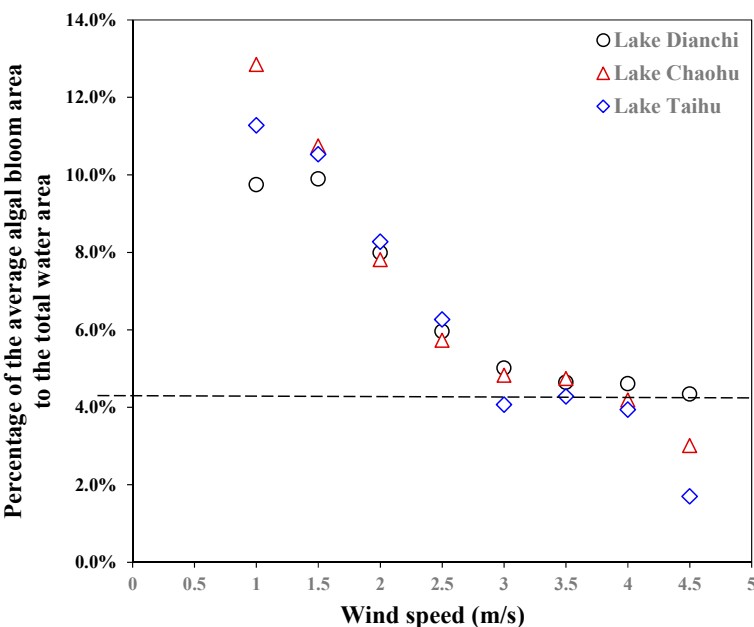

**Figure 6.** Relationship between wind speed and the area of floating algal blooms in Lake Taihu, Lake Chaohu and Lake Dianchi.

Considering the limit of 20% algal bloom area to the total water area for each lake, the maximum wind speed was highest for Lake Dianchi (4.5 m/s), followed by Lake Chaohu (4.0 m/s) and Lake Taihu (3.0 m/s). These maximum wind speeds were obtained considering the full annual cycle of wind speeds and algal blooms. Interestingly, algal blooms in all three lakes had a similar behavior for wind speeds greater than 3 m/s. At this velocity, the average area of algal blooms was 40 % of the total area of the lake surface. In in situ studies of vertical distribution of phytoplankton biomass in Lake Taihu and Lake Chaohu, this same wind speed, 3 m/s, was shown to be a transition point in the near surface mixing regime [14,15,35].

Considering the days in which the 20% algal bloom limit has occurred in the last two decades, it is possible to validate the lake-specific thresholds and the intralake threshold of 3.0 m/s. The number of high-bloom area days that occurred in Lake Taihu when the lake-specific threshold of 3.4 m/s was 86, accounts for 98.8% of the total high bloom events. This percentage is reduced to 94.2% when the intralake threshold of 3.0 m/s is considered. Lake Chaohu had a similar percentage of high-bloom events for its threshold 4.0 m/s (100.0%) and for 3 m/s (97.6%). Lake Dianchi, with the highest lake-specific threshold of 4.5 m/s, had the lowest percentage of high-bloom area days for intralake threshold, 3.0 m/s. The higher number of events beyond intralake threshold are related to this lake's variability, also seen in the higher overall variance of algal bloom dynamics. Shallower water depth is more favorable for cyanobacteria to float to the water surface and form bloom [36], which is also the reason why the wind speed threshold of algae bloom in Lake Taihu is more frequent than the other two lakes.

Spatially extensive algal blooms occur most frequently in the summer months in all three lakes. In a comparison of the relationship between the area of algal blooms and wind speed by month (Figure 7), the results show that in warmer months, the relationship between wind speed and the area of blooms is stronger (higher correlation, $\rho < 0.01$). This was especially true in Lake Taihu and Lake Chaohu, where severe algal blooms occurred more frequently. Nonsignificant or even positive correlations occurred in the coldest months.

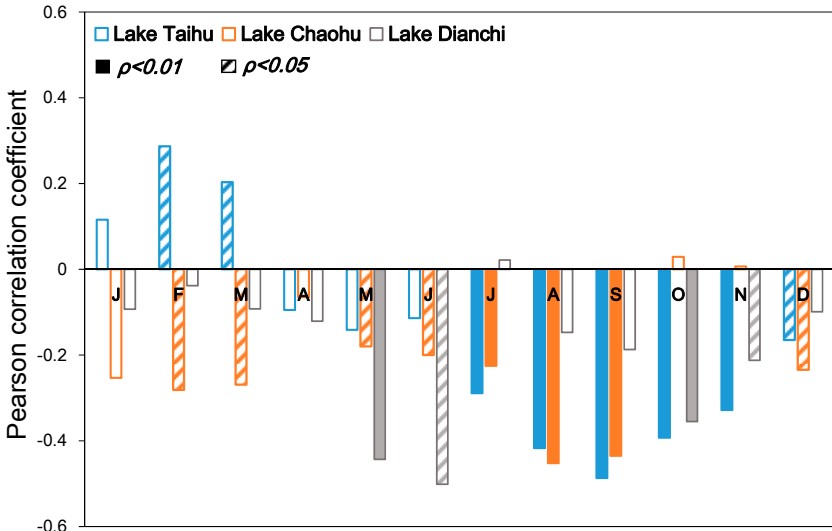

**Figure 7.** Pearson correlation between wind speed and the area of algal blooms by month (indicated by the first letter) for Lake Taihu, Lake Chaohu and Lake Dianchi from 2000 to 2018.

## 4. Discussion

### 4.1. Wind Speed Effect on the Seasonal Cycle of Floating Algal Blooms

Wind speed and wind-speed variance in all three lakes had similar seasonal dynamics (Figure 8). Lake Taihu and Lake Chaohu had two maxima in wind speed, with a rainy season depression during June and July, as well as typhoon-related enhancement in July and August. Accordingly, the algal blooms in Lake Taihu also presented double maxima, even though the first peak was relatively mild. The impact of a wind reduction in the rainy season in Lake Chaohu was much weaker, as severe algal blooms only occurred in summer. Lake Dianchi is located in a different climate area (humid subtropical) and is seldom affected by typhoons; thus, wind speed had a single maximum in late winter and early spring, and a minimum in August, coinciding with the area of maximum algal blooms.

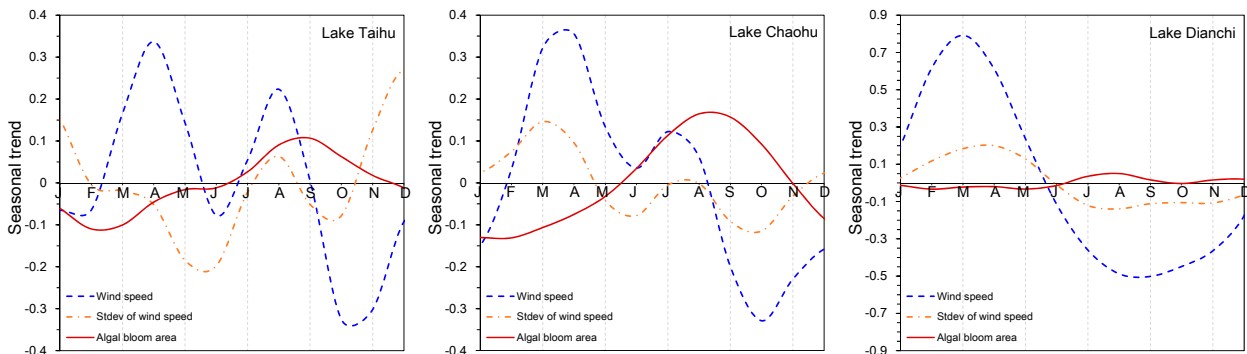

**Figure 8.** Seasonal trends of wind speed, standard deviation of wind speed, and the area of floating algal blooms in Lake Taihu, Lake Chaohu, and Lake Dianchi.

### 4.2. Wind Direction Effects on the Spatial Distribution of Floating Algae Blooms

The dominant wind direction of Lake Chaohu and Lake Taihu was from the east or east-southeast. Floating algae naturally gathered in the downwind direction in these two lakes (Figure 9). However, for Lake Dianchi, the dominant wind direction was from the west-southwest, whereas the highest frequency of floating algae gathering occurred in the western extremes. The divergence field of Lake Dianchi shows that the northern or southern areas were more suitable for floating algae to gather. Likewise, the largest river in the Dianchi catchment flows into the northern end of the lake. It is likely that multiple

divergence/convergence processes determine the spatial distribution of algal blooms in this lake [15]. A comparison of the frequency of floating algae and the divergence field under the average wind speed of dominant wind direction in these three lakes showed that the frequencies of floating algae were negatively correlated with the divergences. Therefore, the dominant wind direction, the lake bathymetry and the river inflow all combine to drive the spatial distribution of floating algae.

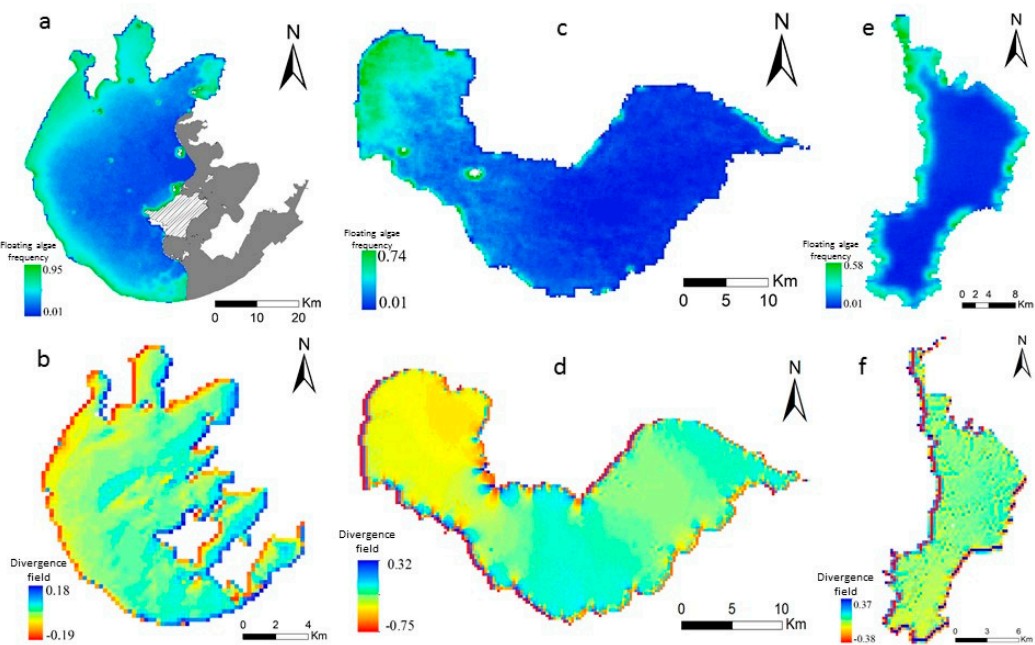

**Figure 9.** The frequency of floating algae and the divergence field under the average wind speed of the dominant wind directions in (**a**,**b**) Lake Taihu; (**c**,**d**) Lake Chaohu; and (**e**,**f**) Lake Dianchi.

### 4.3. Implications for Administrators

As confirmed in the present study, wind speed is a decisive factor in the formation and severity of algae blooms. A significant negative correlation between wind speed and floating algal bloom area was present in all three lakes. Assuming that the wind direction, lake depth, stratification regimes and nutrients are similar, we estimated that conditions will worsen in the coming decades, especially in Lake Taihu. The wind/bloom relationships identified in the present study indicate that there will be a 26% increase in floating algae coverage in Lake Taihu in the coming decades.

Beginning in 2000, the Chinese national and local governments adopted pollution control actions and ecological restoration projects in the catchments of all three lakes [37–39]. The concentrations of nutrients such as nitrogen and phosphorus showed decreasing trends [40], but the average nutrient concentrations still exceed those necessary for controlling algae growth [37,41]. While there is some evidence of improvements in phytoplankton biomass [42,43], it is clear that HABs formation and its related impacts, including the closing of potable water intakes for the cities of Wuxi and Chaohu, will continue, even though the provincial and national governments have made major efforts to reduce eutrophication drivers and restore lake conditions. Lake managers need better tools to better plan such actions.

### 5. Conclusions

The results from the present study indicate that there is a close relationship between local meteorological conditions and the occurrence of major bloom-forming events. For the three lakes examined in the present study, lake-specific thresholds for wind speed could be used as early warning systems for remediation actions to prevent potential impacts

to local catchment communities, both economic and health-related. By linking daily or weekly meteorological predictions to lake conditions, lake managers would be able to act in time. For lakes and reservoirs that share similar characteristics—shallow and extensive waterbodies used for potable water sources—the intralake value of 3 m/s could be used as an initial threshold.

In conclusion, the influence of climate change on wind indirectly affects the occurrence and severity of algal blooms in eutrophic lakes. While this was most evident in the shallow lakes of the present study, further study is required to combine thresholds for lake temperature, lake depth and wind speed to create a comprehensive understanding of climate change effects on the phenological processes for inland waters.

**Author Contributions:** Conceptualization, K.S. and Y.Z.; methodology, Y.Z, T.H. and P.Z.; software, Y.Z.; validation, Y.Z. and M.Z.; formal analysis, Y.Z.; investigation, M.H., Y.J. and L.L.; writing—original draft preparation, Y.Z.; writing—review and editing, S.L.; funding acquisition, K.S. and Y.Z. All authors have read and agreed to the published version of the manuscript.

**Funding:** This work was supported by the National Natural Science Foundation of China (Grant No. 41922005 and No. 41671371), Jiangsu Provincial Key Research and Development Program (BE2019774), the Youth Innovation Promotion Association of Chinese Academy of Sciences (2017365) and the Scientific Instrument Developing Project of the Chinese Academy of Sciences (Grant No. YJKYYQ20200071 and No. YJKYYQ20200048).

**Acknowledgments:** Field data were provided by Scientific Data Sharing Platform for Lake and Watershed, Nanjing Institute of Geography and Limnology, Chinese Academy of Sciences. This collaboration was supported by the ESA/MOST Dragon 5 program. We are also grateful to Xiaoli Shi, Zhen Yang, Ligang Xu and Hai Xu for their providing TN and TP data of Lake Taihu, Lake Chaohu and Lake Dianchi.

**Conflicts of Interest:** The authors declare no conflict of interest.

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
