# Peer review of "Wind Effects for Floating Algae Dynamics in Eutrophic Lakes"

_remotesensing, doi:10.3390/rs13040800_

Round 1

Reviewer 1 Report

The research topic undertaken by Yuchao Zhang and co-authors in the paper "Wind effects for floating algae dynamics in eutrophic lakes" is interesting and important because of permanent threats of algal blooms to the environment and humans. The research is based on a large database and the statistical methods used are appropriate. However, it seems that the scope of the work could include more relevant information and if it is possible more detailed analyzes.

First of all, information on algal blooms, mainly cyanobacteria, in Chinese lakes has already been presented in many previous studies. It seems that it would be worth presenting them to the reader in a more detailed way. For example, the manuscript uses the term "harmful algae blooms (HABs)", but there is no information whether there were cyanobacteria or other algae species in these studied lakes. Were these taxa toxin producing? Another example: the studied lakes are defined as eutrophic and their trophic status as "moderate" or "mild", but there is no information about the range of nutrient concentrations and their differentiation between lakes. Providing these data by authors, would facilitate the comparison of the obtained results as well as final conclusions included in manuscript with other articles on the similar topic.

In addition, although the main problem indicated in the title is adequately addressed, it may be worth to perform additional analyzes taking into account (apart from wind speed), the influence of temperature, precipitation and concentration of nutrients (e.g. variation partitioning analysis)? They would give an answer to the questions to what extent the wind speed modifies the bloom and which of the environmental factors is more important.

A few comments:

Abstract:

I am not sure whether based on a study covering a period of about 20 years and only one parameter (wind), we can talk about the impact of climate change ("The results indicate that climate change has different ..."). Perhaps it would be more correct to use the term "meteorological influence".

The phrase "..... a common effect on the vertical distribution of algae." is incomprehensible. I did not find in the text the results of differentiation of algae along the water column.

Introduction

The introduction presents the issue and the legitimacy of taking up the topic in a very general way. It would be worth focusing more on the discussed issues in the geographic and climatic region in which the research is carried out and refer to the study of these specific lakes.

Figure 1

The font used in the legend is too small.

Table 1

The caption of the table should include the period from which the averaged values ​​come from, such as air temperature and wind speed.

Data and Methods

2.2 Wind dynamics

In this subsection there is information that data related to precipitation and sunshine hours were used in the study. Did these data were taken for analysis?

The results presented in the article cover the period from 2000 to 2018, while the methodology provides 1956-2016 ? How earlier data (from before 1956-2000) were used for the analyzes?

Figure 3

Too small font in the axis subs.

What statistics does the chart (box-plot for each year) show? I would recommend adding a legend.

Discussion

It seems that the charts presented in this section illustrate the correlations and trends resulting from the analyzes and should be moved, along with their descriptions, to section 3: Results.

Conclusion

This section is too long, it seems that it should be shortened and partially moved to the Discussion section. The conclusions should avoid citations.

Author Response

The research topic undertaken by Yuchao Zhang and co-authors in the paper "Wind effects for floating algae dynamics in eutrophic lakes" is interesting and important because of permanent threats of algal blooms to the environment and humans. The research is based on a large database and the statistical methods used are appropriate. However, it seems that the scope of the work could include more relevant information and if it is possible more detailed analyzes. First of all, information on algal blooms, mainly cyanobacteria, in Chinese lakes has already been presented in many previous studies. It seems that it would be worth presenting them to the reader in a more detailed way. For example, the manuscript uses the term "harmful algae blooms (HABs)", but there is no information whether there were cyanobacteria or other algae species in these studied lakes. Were these taxa toxin producing? Response: In the studied lakes, harmful cyanobacteria were the dominant species. Lake Taihu was dominated by Microcystis(Zhang et al., 2018), Lake Chaohu was dominated by Microcystis and Dolichospermum (Zhang, Yang, Yu, & Shi, 2020) and Lake Dianchi was dominated by Microcystis and Aphanizomenon (Shan et al., 2019). All these cyanobacteria were toxin producing taxa (Brand, 1996). Pls see Introduction Section in detail. Brand, L. E. (1996). Algae: An Introduction to Phycology. Bulletin of Marine Science -Miami-, 59(1), 242-243(242). Shan, K., Song, L., Chen, W., Li, L., Liu, L., Wu, Y., . . . Peng, L. (2019). Analysis of environmental drivers influencing interspecific variations and associations among bloom-forming cyanobacteria in large, shallow eutrophic lakes. Harmful Algae, 84, 84-94. doi:https://doi.org/10.1016/j.hal.2019.02.002 Zhang, M., Shi, X., Yang, Z., Yu, Y., Shi, L., & Qin, B. (2018). Long-term dynamics and drivers of phytoplankton biomass in eutrophic Lake Taihu. Science of the Total Environment, 645, 876-886. Zhang, M., Yang, Z., Yu, Y., & Shi, X. (2020). Interannual and Seasonal Shift between Microcystis and Dolichospermum: A 7-Year Investigation in Lake Chaohu, China. Water, 12(7), 1978. Another example: the studied lakes are defined as eutrophic and their trophic status as "moderate" or "mild", but there is no information about the range of nutrient concentrations and their differentiation between lakes. Providing these data by authors, would facilitate the comparison of the obtained results as well as final conclusions included in manuscript with other articles on the similar topic. Response: In the revised version, we added the TN and TP concentrations of these three lakes in Table 1. Pls see Table 1 in detail. In addition, although the main problem indicated in the title is adequately addressed, it may be worth to perform additional analyzes taking into account (apart from wind speed), the influence of temperature, precipitation and concentration of nutrients (e.g. variation partitioning analysis)? They would give an answer to the questions to what extent the wind speed modifies the bloom and which of the environmental factors is more important. Response: The authors agree with the reviewer very much. There is no doubt that the formation of algal blooms is a very complex process with many influencing factors Including temperature, precipitation and nutrient content and so on. Since 2000, the concentration of nutrients in the three lakes has not been fundamentally improved, and these lakes were still in a state of eutrophication. In addition, taking Taihu Lake as an example, the occurrence of algal blooms in winter is not a new case. In February 2020, hundreds of square kilometers of algal blooms occurred. Therefore, we believe that wind is an important factor in the formation of algal blooms by affecting the vertical distribution of cyanobacteria without fundamental improvement of water quality. This research results will provide important support for cyanobacteria bloom prediction and early warning. A few comments: Abstract: I am not sure whether based on a study covering a period of about 20 years and only one parameter (wind), we can talk about the impact of climate change ("The results indicate that climate change has different ..."). Perhaps it would be more correct to use the term "meteorological influence". Response: Thanks for this comment. Recent studies indicate that climate change is expected to reduce wind speed in the eastern Asian atmospheric circulation. Although this study covered only about 20 years, it is still a segment in the process of wind speed mitigation, so we use climate change instead of meteorological impact/ influence. The phrase "..... a common effect on the vertical distribution of algae." is incomprehensible. I did not find in the text the results of differentiation of algae along the water column. Response:Thanks for this comment. We added some references about the wind effects on vertical distributions of the algae in the water column in the Introduction Section.” Surface and vertical distributions of the total algae in shallow lakes were markedly affected by wind, and the coefficients of variation of pigment concentrations throughout the water column showing a negative correlation with increasing wind (Cao et al., 2006; Xue et al., 2015; Zhang et al., 2016)”. Pls see Introduction Section in detail. Introduction The introduction presents the issue and the legitimacy of taking up the topic in a very general way. It would be worth focusing more on the discussed issues in the geographic and climatic region in which the research is carried out and refer to the study of these specific lakes. Figure 1:The font used in the legend is too small. Response: Thanks for this suggestion. We have remade the map to make sure it is clear and accurate. Pls see Figure 1 in detail. Table 1:The caption of the table should include the period from which the averaged values come from, such as air temperature and wind speed. Response:We have revised this caption to “Description of the basic features of Lake Taihu, Lake Chaohu, and Lake Dianchi during 2000-2018”. Pls see Table 1 in detail. Data and Methods 2.2 Wind dynamics In this subsection there is information that data related to precipitation and sunshine hours were used in the study. Did these data were taken for analysis? Response: In order to achieve wind direction effects on the spatial distribution of floating algae blooms, we used the Princeton Ocean Model to simulate the algae distribution under average wind speed and dominant wind direction. Daily data (air temperature, wind direction, wind speed, precipitation, sunshine hours etc.) were obtained for the nearest national weather stations (i.e., Dongshan, Hefei and Kunming Stations) from 2000 to 2018 from the China Meteorological Data Sharing Service System (http://cdc.cma.gov.cn/home.do). Pls see section 2.2 in detail. The results presented in the article cover the period from 2000 to 2018, while the methodology provides 1956-2016 ? How earlier data (from before 1956-2000) were used for the analyzes? Response: Although we have the wind speed data since 1956, which is helpful to understand the wind speed trend since 1956, the algal bloom data started from 2000. Therefore, our research is based on the algal bloom data and synchronous wind speed and direction data from 2000 to 2018. Pls see sections 2.1 & 2.2. Figure 3 Too small font in the axis subs. What statistics does the chart (box-plot for each year) show? I would recommend adding a legend. Response: We revised this figure due to this suggestion. Pls see Figure 3 in detail. Discussion It seems that the charts presented in this section illustrate the correlations and trends resulting from the analyzes and should be moved, along with their descriptions, to section 3: Results. Response: Thanks for this suggestion. We have moved former section 4.1 to Results Section. Pls see Result Section in detail. Conclusion This section is too long, it seems that it should be shortened and partially moved to the Discussion section. The conclusions should avoid citations. Response: Thanks for this suggestion. We have shortened and reorganized Conclusion Section. We added a new section “implications for administrators” Discussion Section and moved part of the conclusion into it. Pls see Section 4.3 and 5 in detail.

Reviewer 2 Report

The authors undertook the important task of assessing the relationship between the influence of meteorological conditions on the formation and intensity of algal blooms in 3 eutrophic shallow lakes in China. The studies were well designed, written and referenced. The figures and tables are adequate. The manuscript could be accepted for publishing after minor revision.

Detail comment:

Figure 1 -  Lake names or symbols should be added. Not everyone knows the location of the lakes.

Author Response

Reviewer 2:

The authors undertook the important task of assessing the relationship between the influence of meteorological conditions on the formation and intensity of algal blooms in 3 eutrophic shallow lakes in China. The studies were well designed, written and referenced. The figures and tables are adequate. The manuscript could be accepted for publishing after minor revision.

Detail comment:

Figure 1 -  Lake names or symbols should be added. Not everyone knows the location of the lakes.

Response: Thanks for this suggestions. We have added the lake names into this figure in the revised version.

Pls see Figure 1 in detail.

Reviewer 3 Report

Please find enclosed my comments

Author Response

Reviewer 3:

This paper is aimed to assess the influence of the wind in algae biomass distribution of three typical eutrophic lakes in China: 17 Lake Taihu, Lake Chaohu and Lake Dianchi. The authors used a Princeton Ocean Model to determine wind conditions above the study areas and a software named BFAST to estimate phonological changes in the ecosystems, using time series satellite images. They decomposed the time series of algal blooms and wind dynamics into trend, and used Durbin-Watson analysis to identify autocorrelation of the residual components. The authors were able to determine the critical threshold values of the wind speed for the setup of the chlorophyll-a concentrations blooms, as well as significant correlation between wind speed and floating algal bloom, which they found it were negatives, as well as to identify locations that were particularly susceptible to algae blooms. They concluded that algal bloom conditions will worsen in the coming decades due to the continuous decline of the wind. The paper deals with a very important and current subject and issue, that is, the setup of Harmful algae blooms (HABs), which put is risk the health of the inhabitant of several aquatic systems across the globe. I found the paper interesting covering an important topic on the ecological restoration the studied lakes, as well for several other catchment areas throughout China. It was easy to read, the English language sounds correct. I, therefore, recommend the publication of this manuscript after minor revision/corrections, among those I suggested below.

General remarks:

-Please better explain the relationship between the wind divergence and the implication for the convergence of the algae blooms

Response: We give a helpful and interesting explanation to the divergence. Image several current vectors  with different lengths and directions shown in Fig.1. They will through a control volume . Some vectors may through it toward outside (,,), but others may through it toward inside (,). According to the Green’s theorem, integrated the divergence over the control volume , we can transfer the surface integral to the line integral, and the term on the right side is the net flow flux along the interface of control volume. That is, the divergence is linked with net flow flux.

Figure 1

As to the algal colonies which suspended in water, they should move followed by current. So there is also a vivid metaphor: If the net flow flux is larger than 0, it indicates the algal colony is out of this control volume. The larger the flux is, the faster the departure speed is. That is divergence, the bloom is difficult to form. On the contrary, if the net flow flux is less than 0, it indicates the algal colony doesn’t want to leave the control volume. It becomes very slow, but it still will leave. That is convergence, the blooms are easy to form.

Reference:

Li W, Qin BQ. Dynamics of spatiotemporal heterogeneity of cyanobacterial blooms in large eutrophic Lake Taihu, China. Hydrobiologia, 2019, 833: 81-93.

Pls see Section 2.2 in detail.

- Better explain BFAST and Durbin-Watson analysis. Add references.

Response: A Breaks For Additive Seasonal and Trend (BFAST) was used to decompose each time series, algal blooms and wind dynamics into trend, seasonal, and residual (irregular) components. Many previous studies have shown that the BFAST method can accurately detect the long-term trend of phenological changes and abrupt changes in multi-temporal satellite images (Verbesselt et al., 2010; Xue et al., 2014). Chen et al. used the BFAST method to detect the time and magnitude of the abrupt change in the NDVI (Normalized Difference Vegetation Index) trend component, and finally revealed the spatio-temporal changes of land cover in the wetland ecosystem of Poyang Lake (Chen et al.,2014). In addition, the main advantage of BFAST is its ability to detect breakpoints in the linear trend. Many studies found that the decomposition of the NDVI and EVI (Enhanced Vegetation Index) time series based on BFAST method can effectively detect the impact on vegetation growth of exceptional events, such as floods and fires (Lambert et al., 2013; Watts et al., 2014; Fang et al., 2018; Geng et al., 2019). The BFAST method is also applied in identifying newly dammed reservoirs from time series of MODIS-derived NDWI (Normalized Difference Water Index) images (Zhang et al., 2018; Deng et al., 2020).

Durbin-Watson analysis was written by mistake. We have deleted it in the revised version.

Pls see Section 2.3 in detail.

References:

Verbesselt J, Hyndman R, Zeileis A, et al. Phenological change detection while accounting for abrupt and gradual trends in satellite image time series[J]. Remote Sensing of Environment, 2010, 114(12): 2970-2980.

Xue Z, Du P, Feng L.Phenology-Driven Land Cover Classification and Trend Analysis Based on Long-term Remote Sensing Image Series[J].IEEE Journal of Selected Topics in Applied Earth Observations and Remote Sensing, vol. 7, no. 4, pp. 1142-1156, April 2014, doi: 10.1109/JSTARS.2013.2294956.

Chen L, Michishita R, Xu B. Abrupt spatiotemporal land and water changes and their potential drivers in Poyang Lake, 2000–2012[J]. ISPRS journal of photogrammetry and remote sensing, 2014, 98: 85-93.

Lambert J, Drenou C, Denux J P, et al. Monitoring forest decline through remote sensing time series analysis[J]. GIScience & Remote Sensing, 2013, 50(4): 437-457.

Watts L M, Laffan S W. Effectiveness of the BFAST algorithm for detecting vegetation response patterns in a semi-arid region[J]. Remote Sensing of Environment, 2014, 154: 234-245.

Fang X, Zhu Q, Ren L, et al. Large-scale detection of vegetation dynamics and their potential drivers using MODIS images and BFAST: A case study in Quebec, Canada[J]. Remote Sensing of Environment, 2018, 206: 391-402.

Geng L, Che T, Wang X, et al. Detecting spatiotemporal changes in vegetation with the BFAST model in the Qilian Mountain region during 2000–2017[J]. Remote Sensing, 2019, 11(2): 103.

Zhang, W., et al., 2018. Identifying emerging reservoirs along regulated rivers using multi-source remote sensing observations. Remote Sens. 11 (1), 25.

Deng X, Song C, Liu K, et al. Remote sensing estimation of catchment-scale reservoir water impoundment in the upper Yellow River and implications for river discharge alteration[J]. Journal of Hydrology, 2020, 585: 124791.

- Please revise the sentence

“These minimum wind speeds considered the full annual cycle of wind speeds and algal blooms”

Response: We have revised this sentence to “These maximum wind speeds were obtained considering the full annual cycle of wind sp

Reviewer 3:

This paper is aimed to assess the influence of the wind in algae biomass distribution of three typical eutrophic lakes in China: 17 Lake Taihu, Lake Chaohu and Lake Dianchi. The authors used a Princeton Ocean Model to determine wind conditions above the study areas and a software named BFAST to estimate phonological changes in the ecosystems, using time series satellite images. They decomposed the time series of algal blooms and wind dynamics into trend, and used Durbin-Watson analysis to identify autocorrelation of the residual components. The authors were able to determine the critical threshold values of the wind speed for the setup of the chlorophyll-a concentrations blooms, as well as significant correlation between wind speed and floating algal bloom, which they found it were negatives, as well as to identify locations that were particularly susceptible to algae blooms. They concluded that algal bloom conditions will worsen in the coming decades due to the continuous decline of the wind. The paper deals with a very important and current subject and issue, that is, the setup of Harmful algae blooms (HABs), which put is risk the health of the inhabitant of several aquatic systems across the globe. I found the paper interesting covering an important topic on the ecological restoration the studied lakes, as well for several other catchment areas throughout China. It was easy to read, the English language sounds correct. I, therefore, recommend the publication of this manuscript after minor revision/corrections, among those I suggested below.

General remarks:

-Please better explain the relationship between the wind divergence and the implication for the convergence of the algae blooms

Response: We give a helpful and interesting explanation to the divergence. Image several current vectors  with different lengths and directions shown in Fig.1. They will through a control volume . Some vectors may through it toward outside (,,), but others may through it toward inside (,). According to the Green’s theorem, integrated the divergence over the control volume , we can transfer the surface integral to the line integral, and the term on the right side is the net flow flux along the interface of control volume. That is, the divergence is linked with net flow flux.

Figure 1

As to the algal colonies which suspended in water, they should move followed by current. So there is also a vivid metaphor: If the net flow flux is larger than 0, it indicates the algal colony is out of this control volume. The larger the flux is, the faster the departure speed is. That is divergence, the bloom is difficult to form. On the contrary, if the net flow flux is less than 0, it indicates the algal colony doesn’t want to leave the control volume. It becomes very slow, but it still will leave. That is convergence, the blooms are easy to form.

Reference:

Li W, Qin BQ. Dynamics of spatiotemporal heterogeneity of cyanobacterial blooms in large eutrophic Lake Taihu, China. Hydrobiologia, 2019, 833: 81-93.

Pls see Section 2.2 in detail.

- Better explain BFAST and Durbin-Watson analysis. Add references.

Response: A Breaks For Additive Seasonal and Trend (BFAST) was used to decompose each time series, algal blooms and wind dynamics into trend, seasonal, and residual (irregular) components. Many previous studies have shown that the BFAST method can accurately detect the long-term trend of phenological changes and abrupt changes in multi-temporal satellite images (Verbesselt et al., 2010; Xue et al., 2014). Chen et al. used the BFAST method to detect the time and magnitude of the abrupt change in the NDVI (Normalized Difference Vegetation Index) trend component, and finally revealed the spatio-temporal changes of land cover in the wetland ecosystem of Poyang Lake (Chen et al.,2014). In addition, the main advantage of BFAST is its ability to detect breakpoints in the linear trend. Many studies found that the decomposition of the NDVI and EVI (Enhanced Vegetation Index) time series based on BFAST method can effectively detect the impact on vegetation growth of exceptional events, such as floods and fires (Lambert et al., 2013; Watts et al., 2014; Fang et al., 2018; Geng et al., 2019). The BFAST method is also applied in identifying newly dammed reservoirs from time series of MODIS-derived NDWI (Normalized Difference Water Index) images (Zhang et al., 2018; Deng et al., 2020).

Durbin-Watson analysis was written by mistake. We have deleted it in the revised version.

Pls see Section 2.3 in detail.

References:

Verbesselt J, Hyndman R, Zeileis A, et al. Phenological change detection while accounting for abrupt and gradual trends in satellite image time series[J]. Remote Sensing of Environment, 2010, 114(12): 2970-2980.

Xue Z, Du P, Feng L.Phenology-Driven Land Cover Classification and Trend Analysis Based on Long-term Remote Sensing Image Series[J].IEEE Journal of Selected Topics in Applied Earth Observations and Remote Sensing, vol. 7, no. 4, pp. 1142-1156, April 2014, doi: 10.1109/JSTARS.2013.2294956.

Chen L, Michishita R, Xu B. Abrupt spatiotemporal land and water changes and their potential drivers in Poyang Lake, 2000–2012[J]. ISPRS journal of photogrammetry and remote sensing, 2014, 98: 85-93.

Lambert J, Drenou C, Denux J P, et al. Monitoring forest decline through remote sensing time series analysis[J]. GIScience & Remote Sensing, 2013, 50(4): 437-457.

Watts L M, Laffan S W. Effectiveness of the BFAST algorithm for detecting vegetation response patterns in a semi-arid region[J]. Remote Sensing of Environment, 2014, 154: 234-245.

Fang X, Zhu Q, Ren L, et al. Large-scale detection of vegetation dynamics and their potential drivers using MODIS images and BFAST: A case study in Quebec, Canada[J]. Remote Sensing of Environment, 2018, 206: 391-402.

Geng L, Che T, Wang X, et al. Detecting spatiotemporal changes in vegetation with the BFAST model in the Qilian Mountain region during 2000–2017[J]. Remote Sensing, 2019, 11(2): 103.

Zhang, W., et al., 2018. Identifying emerging reservoirs along regulated rivers using multi-source remote sensing observations. Remote Sens. 11 (1), 25.

Deng X, Song C, Liu K, et al. Remote sensing estimation of catchment-scale reservoir water impoundment in the upper Yellow River and implications for river discharge alteration[J]. Journal of Hydrology, 2020, 585: 124791.

- Please revise the sentence

“These minimum wind speeds considered the full annual cycle of wind speeds and algal blooms”

Response: We have revised this sentence to “These maximum wind speeds were obtained considering the full annual cycle of wind speeds and algal blooms”.

Pls see last third paragraph in Section Results in detail.

eeds and algal blooms”.

Pls see last third paragraph in Section Results in detail.